# Prognostic Markers of Survival among Japanese Patients with Anaplastic Lymphoma Kinase-Positive Non-Small-Cell Lung Cancer Receiving First-Line Alectinib

**DOI:** 10.3390/diagnostics11122170

**Published:** 2021-11-23

**Authors:** Takayuki Takeda, Tadaaki Yamada, Keiko Tanimura, Takayuki Nakano, Masaki Ishida, Yusuke Tachibana, Shinsuke Shiotsu, Shigeto Horiuchi, Makoto Hibino, Asuka Okada, Yusuke Chihara, Koichi Takayama

**Affiliations:** 1Department of Respiratory Medicine, Japanese Red Cross Kyoto Daini Hospital, Kyoto 602-8026, Japan; keiko-t@koto.kpu-m.ac.jp (K.T.); tnakano@koto.kpu-m.ac.jp (T.N.); 2Department of Pulmonary Medicine, Graduate School of Medical Science, Kyoto Prefectural University of Medicine, Kyoto 602-8566, Japan; tayamada@koto.kpu-m.ac.jp (T.Y.); mishida@koto.kpu-m.ac.jp (M.I.); takayama@koto.kpu-m.ac.jp (K.T.); 3Department of Respiratory Medicine, Japanese Red Cross Kyoto Daiichi Hospital, Kyoto 605-0981, Japan; yutachib@koto.kpu-m.ac.jp (Y.T.); sshiotsu@gmail.com (S.S.); 4Department of Respiratory Medicine, Shonan Fujisawa Tokushukai Hospital, Fujisawa 251-0041, Japan; Horiuchishigeto@hotmail.co.jp (S.H.); m-hibino@ctmc.jp (M.H.); 5Department of Respiratory Medicine, Saiseikai Suita Hospital, Suita 564-0013, Japan; aska_517@icloud.com; 6Department of Respiratory Medicine, Uji-Tokushukai Medical Center, Uji 611-0041, Japan; c1981311@koto.kpu-m.ac.jp

**Keywords:** alectinib, *anaplastic lymphoma kinase* (*ALK*), non-small-cell lung cancer, platelet-to-lymphocyte ratio (PLR), systemic immune-inflammation index (SII)

## Abstract

The prognoses of patients with non-small-cell lung cancer (NSCLC) harboring *anaplastic lymphoma kinase* (*ALK*) gene rearrangement have dramatically improved with the use of ALK tyrosine kinase inhibitors. Although immunological and nutritional markers have been investigated to predict outcomes in patients with several cancers, their usefulness in targeted therapies is scarce, and their significance has never been reported in patients receiving first-line treatment with alectinib. Meanwhile, neutrophil-to-lymphocyte ratio and platelet-to-lymphocyte ratio (PLR) has been investigated during crizotinib treatment. This multicenter retrospective study evaluated 42 consecutive Japanese patients with *ALK*-positive NSCLC who received first-line treatment with alectinib. Immunological and nutritional markers were evaluated at baseline and 3 weeks after alectinib introduction, and their significance in predicting progression-free survival (PFS) was explored. PFS duration was significantly associated with baseline PLR (hazard ratio (HR): 2.49, *p* = 0.0473), systemic immune-inflammation index (SII; HR: 2.65, *p* = 0.0337), prognostic nutrition index (PNI; HR: 4.15, *p* = 0.00185), and the 3-week values for SII (HR: 2.85, *p* = 0.0473) and PNI (HR: 3.04, *p* = 0.0125). Immunological and nutritional markers could be useful in predicting the outcomes of first-line treatment with alectinib. Since PLR and SII consist of platelet counts, platelet count could be an important constituent of these markers.

## 1. Introduction

Lung cancer remains the leading cause of mortality in cancer patients, and non-small-cell lung cancer (NSCLC) accounts for approximately 80% of all lung cancers [1]. The discovery of oncogenic driver genes and introduction of appropriate tyrosine kinase inhibitors (TKIs) has revolutionized the treatment strategy and outcomes of relevant NSCLC with oncogenic driver genes [2,3]. *Anaplastic lymphoma kinase* (*ALK*) gene rearrangement accounts for 3–5% of NSCLCs. It is a known, strong oncogenic driver, and *echinoderm microtubule*-*associated protein*-*like 4*-*ALK* (*EML4*-*ALK*) is the most common [4]. Crizotinib was the first ALK-TKI to show superior efficacy over platinum-based chemotherapy [5]. Thereafter, second-generation ALK-TKIs, consisting of ceritinib, alectinib, and brigatinib, and third-generation lorlatinib, were developed. In the first-line setting, ceritinib demonstrated superior efficacy over platinum-based chemotherapy [6], while alectinib [7,8], brigatinib [9], and lorlatinib [10] confirmed longer progression-free survival (PFS) than crizotinib.

Alectinib demonstrated superior overall survival (OS) and PFS and less toxicity compared to crizotinib in two randomized phase III trials [7,8]. The lower toxic property of alectinib made it suitable for *ALK*-positive NSCLC patients with poor Eastern Cooperative Oncology Group performance status (ECOG-PS) [11]. However, some patients experienced early progressive disease (PD) during treatment with alectinib [12]. Therefore, it would be useful to identify predictive markers for alectinib, which could help identify patients who would benefit from alectinib continuation and patients who should be switched to second-line treatment.

Immunological and nutritional markers have been investigated as candidate biomarkers predicting OS in patients with several types of cancers. This theory is based on the close relationship between the efficacy of cancer treatment and local or systemic host–tumor interactions, which depend on the patient’s immunological and nutritional status. Candidate predictive markers include the neutrophil-to-lymphocyte ratio (NLR) [13,14,15], platelet-to-lymphocyte ratio (PLR) [16,17,18], prognostic nutrition index (PNI) [19,20], advanced lung cancer inflammation index (ALI) [21,22], and systemic immune-inflammation index (SII) [23,24,25].

However, previous reports on the usefulness of these markers during targeted therapies are scarce. The usefulness of NLR and PLR has been reported in epidermal growth factor receptor-TKI treatment [26,27]. In contrast, PLR could reportedly be a predictive marker in patients with *ALK*-positive NSCLC [28]. Furthermore, NLR and PLR can predict outcomes during crizotinib treatment [29]. However, the investigation of these markers during first-line alectinib treatment has not been explored. To our knowledge, this is the first study to investigate immunological and nutritional markers in first-line alectinib treatment. Considering how less toxic the profile of alectinib is, compared to crizotinib, the association between these markers and outcomes during alectinib treatment could differ from those observed during crizotinib treatment. Therefore, we aimed to identify the predictors of outcomes during first-line alectinib treatment for *ALK*-positive NSCLC and examined candidate markers at baseline and mid-treatment.

## 2. Patients and Methods

### 2.1. Patients

This multicenter retrospective study evaluated consecutively 42 Japanese patients with metastatic or recurrent *ALK*-positive NSCLC who received first-line alectinib at six hospitals in Japan between 1 September 2015, and 31 March 2021. The *ALK* rearrangement was diagnosed by immunohistochemistry (IHC) in 11 patients, both IHC and fluorescence in situ hybridization (FISH) in 27, FISH in 1, reverse transcriptase-polymerase chain reaction (RT-PCR) in 1, both IHC and RT-PCR in 1, and next-generation sequencing (Oncomine Dx Target Test Multi-CDx system, Thermo Fisher Scientific Life Technologies Corporation Japan, Tokyo, Japan) in 1. All patients received oral alectinib at a dose of 300 mg twice daily, in compliance with the authorized dosage in Japan. The study protocol complied with the Declaration of Helsinki and was approved by the Ethics Committee of the Japanese Red Cross Kyoto Daini Hospital (4 June 2021; S2021-09) and of each participating hospital. The requirement for informed consent was waived as this was a retrospective analysis of anonymized patient data. Patients were allowed to opt out of the research’s use of their data, and related information is publicly available on each hospital’s website.

### 2.2. Data Collection

Baseline characteristics (age, sex, smoking history, ECOG-PS score, height, body weight, weight loss of ≥5% 6 months before alectinib initiation, body mass index (BMI), histology, blood examination, and the efficacy of alectinib, including objective response and PFS) were collected from electronic medical records.

### 2.3. Study Variables

Previous reports were used to define cut-off values for the following immunological and nutritional markers: NLR, PLR, SII, ALI, and PNI. The NLR was calculated as the ratio of absolute neutrophil count (/µL) to lymphocyte count (/µL), and patients were grouped according to two cut-off values for NLR < 5 or ≥5, which is usually adopted in studies, and <3 or ≥3, which was adopted in the study dealing with *ALK*-positive NSCLC [28]. The PLR was calculated as the ratio of absolute platelet count (/µL) divided by absolute lymphocyte count (/µL) and grouped according to PLR values of <250 or ≥250. The cut-off value for PLR was set at <250 or ≥250 according to the previous report adopting <246.36 or ≥246.36 [18]. The SII values were calculated as absolute platelet count (/L) × NLR, and patients were grouped according to SII values of <1000 or ≥1000. The cut-off values for PLR and SII were originally set based on the receiver operating characteristic curve, based on data of the current study participants. ALI was calculated as (BMI × albumin (g/dL))/NLR, and patients were grouped according to ALI values of <18 or ≥18. The PNI was calculated as 10 × albumin (g/dL) + 0.005 × absolute lymphocyte count (/µL), and patients were grouped according to PNI values of ≥40 or <40.

These markers were evaluated at baseline and 3 weeks after alectinib initiation, for associations with PFS at these two time points. Patients who experienced PD within 3 weeks were excluded from subsequent time point analyses.

After the evaluation of candidate markers at baseline, the composite markers were evaluated, combining one nutritional marker (ALI or PNI) and one immunological marker (NLR, PLR, or SII). At this time, the combination of either of the nutritional markers and the statistically significant immunological markers, as well as the combination of 2 statistically significant immunological markers were evaluated.

### 2.4. Assessments

Tumor response to treatment was assessed using the Response Evaluation Criteria in Solid Tumors (version 1.1). Chest radiography and computed tomography were used to evaluate treatment response, which was determined by the treating physicians and radiologists.

### 2.5. Statistical Analysis

Statistical analyses were performed using EZR [30], a graphical user interface for R software (The R Foundation for Statistical Computing, Vienna, Austria). Baseline characteristics were compared using Pearson’s chi-squared test. The median PFS interval, with corresponding 95% confidence intervals (CIs) and an objective response rate (ORR), was calculated. Curves for PFS were evaluated using the Kaplan–Meier method and log-rank test. Univariate Cox proportional hazards regression analyses were performed for each potential marker, although multivariate analyses were not performed because of the small sample size. Differences were considered statistically significant at *p* < 0.05.

## 3. Results

### 3.1. Baseline Patient Characteristics

As shown in Table 1, there were 42 patients (20 women (47.6%) and 22 men (52.4%)) with a median age of 67 years (range: 29–85 years), of which 37 (88.1%) had a PS of 0 or 1, three (7.1%) a PS of 2, and two (4.8%) a PS of 3. Twelve patients (28.6%) had never smoked. Median BMI was 21.9 (range: 15.3–33.5), and 12 patients (28.6%) had weight loss of ≥5% within 6 months prior to alectinib initiation. The histological types were adenocarcinoma in 40 patients (95.2%) and NSCLC-not otherwise specified in two (4.8%). The sites of metastases at baseline were liver in 7 (16.7%), brain in 11 (26.2%), bone in 13 (31.0%), and pleural dissemination in 22 (52.4%); 21 patients (50.0%) exhibited more than one site of metastasis. The number of patients distributed according to cut-off values for NLR, PLR, SII, ALI, and PNI at baseline and 3 weeks after alectinib initiation is shown in Table 1.

### 3.2. ORR and PFS Outcomes

The treatment responses evaluated after 3 weeks of alectinib treatment were classified as complete response in 5 (11.9%), partial response in 31 (73.8%), stable disease in 2 (4.8%), and PD in 4 (9.5%). The median PFS was 44.2 months (95% CI: 14.6 months, not reached). The Kaplan–Meier curve for PFS in all patients is shown in Figure 1.

### 3.3. Relationships between the Baseline Characteristics or Candidate Predictive Biomarkers and PFS

Univariate analyses were performed to predict PFS for baseline characteristics and the markers at baseline and 3 weeks after alectinib initiation (Table 2). PFS outcomes were significantly associated with baseline ECOG-PS (hazard ratio (HR): 3.75, *p* = 0.0300), PLR (HR: 2.49, *p* = 0.0473), SII (HR: 2.65, *p* = 0.0337), and PNI (HR: 4.15, *p* = 0.00185), as well as 3-week values for SII (HR: 2.85, *p* = 0.0473) and PNI (HR: 3.04, *p* = 0.0125).

Figure 2 shows clear differences in the Kaplan–Meier curves for PFS according to the predictive markers at baseline, with statistical significance from the univariate analyses: the baseline PLR (Figure 2A), SII (Figure 2B), and PNI (Figure 2C). Figure 3 shows the same differences according to the markers from the univariate analyses after 3 weeks: the 3-week values for SII (Figure 3A) and PNI (Figure 3B).

After the evaluation of candidate markers at baseline, the composite markers combining one of the nutritional markers (ALI or PNI) and the statistically significant immunological markers (PLR or SII), as well as the composite of significant immunological markers (PLR and SII), were evaluated. Namely, the following five comparisons were added: (1) “ALI ≥ 18 and PLR < 250” vs. “ALI < 18 and/or PLR ≥ 250”, (2) “ALI ≥ 18 and SII < 1000” vs. “ALI < 18 and/or SII ≥ 1000”, (3) “PNI ≥ 40 and PLR < 250” vs. “PNI < 40 and/or PLR ≥ 250”, (4) “PNI ≥ 40 and SII < 1000” vs. “PNI < 40 and/or SII ≥ 1000”, and (5) “PLR < 250 and SII < 1000” vs. “PLR ≥ 250 and/or SII ≥ 1000”. Although the additive effects were expected through the process, the statistically significant findings were observed in (4) and (5). *p*-values for the abovementioned five groups were 0.475 (1), 0.365 (2), 0.181 (3), 0.036 (4), and 0.010 (5), respectively. The Kaplan–Meier curves of PFS according to the two composite markers with significance are shown in Figure 4: “PNI ≥ 40 and SII < 1000” vs. “PNI < 40 and/or SII ≥ 1000” (Figure 4A) and “PLR < 250 and SII < 1000” vs. “PLR ≥ 250 and/or SII ≥ 1000” (Figure 4B).

### 3.4. Adverse Events (AEs)

Any grade AEs were observed in 10 patients (23.8%); grade 3–4 AEs were not observed. Renal dysfunction was observed in four patients (9.5%: grade 1, n = 3; grade 2, n = 1), edema in two (4.8%: grade 1, n = 2), and peripheral neuropathy in two (4.8%: grade 1, n = 2). Anemia, bradycardia, skin toxicity (2.4%: grade 1), and liver dysfunction (2.4%: grade 2) were each observed in one patient. The grade 2 renal dysfunction was diagnosed as transient dehydration. All cases recovered without dose reduction or cessation (Table 3).

## 4. Discussion

The discovery of *ALK* gene rearrangement and the introduction of ALK-TKIs have substantially improved the prognosis of patients with *ALK*-positive NSCLC. However, patients treated with ALK-TKIs ultimately face resistance. The resistance mechanisms have been vigorously investigated. Mutations in target tyrosine kinases (second-site gatekeeper mutation), activation of alternative pathways, epithelial-to-mesenchymal transition, phenotypical changes, and drug transporter activation are the major mechanisms [31]. The different resistance mechanisms in first- and second-generation ALK-TKIs have been documented in terms of second-site gatekeeper mutations [32].

Meanwhile, biomarkers to predict outcomes of ALK-TKIs are limited. B cell lymphoma-2-like protein 11 (known as BIM) deletion polymorphism [33] and differential protein stability [34] are related to the outcomes of ALK-TKIs on a molecular basis, while PLR can predict the prognosis of patients with *ALK*-positive NSCLC [28]. NLR and PLR are associated with the outcomes of crizotinib treatment on a practical basis [29]. Immunological and nutritional markers are suitable for repetitive evaluation, which reflects dynamically changing host–tumor interactions. Considering the difference in therapeutic effects, and adverse events between crizotinib and alectionib, it is expected that the biomarkers could be different between these ALK-TKIs.

Contrary to our prospect of the different biomarker profiles between them, baseline PLR and SII, both comprising platelet count, were significantly associated with the outcomes of alectinib treatment in the current study. This finding was consistent with previous reports on crizotinib treatment [29], although alectinib had different effects on the efficacy and adverse events of crizotinib [7,8,11]. The observed findings suggest that platelet count could influence the prognosis of patients with *ALK*-positive NSCLC.

Platelets are considered favorable in tumor spread by stimulating cancer cell proliferation and facilitating metastasis [35]. Considering the relationship between *ALK*-positive NSCLC and platelets, *EML4*-*ALK* rearrangements were found in platelets. Their detection was associated with poor outcomes during crizotinib treatment [36]. This suggests that tumor-educated platelets play a key role and could be predictive markers during the treatment of *ALK*-positive NSCLC.

Platelet-derived growth factors (PDGFs) and their receptors (PDGFRs) have been found to correlate with tumor progression. Phosphorylated stem cell growth factor receptor kit and PDGF-α have been reported as predictive and prognostic factors in *ALK*-positive NSCLC [37]. Platelets contain several growth factors, including PDGFs and transforming growth factor-β (TGF-β), which favor tumor progression [38]. Platelet counts are regulated by thrombopoietin, which has close interaction with PDGFs and TGF-β [39]. The fact that *ALK*-positive NSCLC has a close relationship with platelets and PDGFR-α could explain the observed outcome in biomarkers composed of platelet count and the outcomes of alectinib treatment.

Conversely, *ALK*-positive NSCLC is associated with a high rate of venous thromboembolism [40]. Platelets have been found to have a closer relationship with venous thromboembolism than previously anticipated [41]. Thus, the role of platelets in *ALK*-positive NSCLC could be of clinical importance, which reinforces the aforementioned mechanism of platelet and platelet-containing PDGFs and TGF-β.

Regarding the importance of the host’s nutritional status, the current study demonstrated the efficacy of baseline ECOG-PS and PNI, as well as 3-week values of PNI in predicting PFS. This result suggests the importance of the baseline nutritional status during alectinib treatment.

The timing of the evaluation is another scientific interest. Our results demonstrated the importance of baseline PLR, SII, and PNI values. Contrary to our expectations, the immunological and nutritional markers did not change in favor of better outcomes after 3-week treatment with alectinib, which is considered a sufficient treatment period for the efficacy of alectinib in tumors. PLR was the only marker that lost significance after 3 weeks of alectinib treatment. These observations suggest that the immunological and nutritional status of *ALK*-positive NSCLC does not change dramatically after alectinib treatment, irrespective of treatment efficacy. Thus, the baseline values of PLR, SII, and PNI could be of the utmost importance in predicting outcomes.

The current study has several limitations. Firstly, a retrospective design is prone to bias. Secondly, the small sample size due to the rarity of *ALK*-positive NSCLC is another potential source of bias. The small sample size hampered multivariate analyses. Multivariate analyses would have allowed us to investigate the significance of SII and PNI, since these two markers are closely related to ECOG-PS. Thirdly, we selected the candidate immunological and nutritional markers based on the results of the original articles. For example, we did not adopt lung immune prognostic index (LIPI), because LIPI was associated with the outcomes of immune checkpoint inhibitors but not with chemotherapy [42]. Therefore, there could be some potential markers, which we did not analyze. Finally, we selected cut-off values for the various predictive markers from previous reports, save for PLR and SII. Therefore, the utility of markers and optimal cut-off values should be clarified in large prospective studies.

## 5. Conclusions

The present study revealed that the PFS of Japanese patients with *ALK*-positive NSCLC who received first-line treatment with alectinib could be related to immunological and nutritional status. PLR and SII at baseline and SII after 3 weeks were significantly associated with PFS. Since these markers consist of platelet counts, the current study suggests the importance of platelet count as a constituent of immunological markers in predicting the outcome of *ALK*-positive NSCLC. Baseline ECOG-PS and PNI are also important in predicting PFS, which reflect the nutritional status of the host.

## Figures and Tables

**Figure 1 diagnostics-11-02170-f001:**
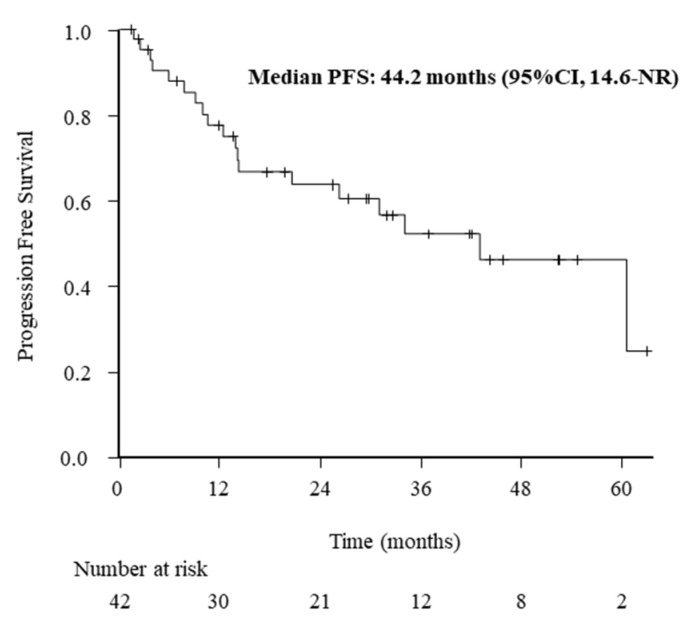
Kaplan–Meier curve of progression-free survival (PFS) in all patients. The median PFS was 44.2 months (95% confidence interval: 14.6 months, not reached).

**Figure 2 diagnostics-11-02170-f002:**
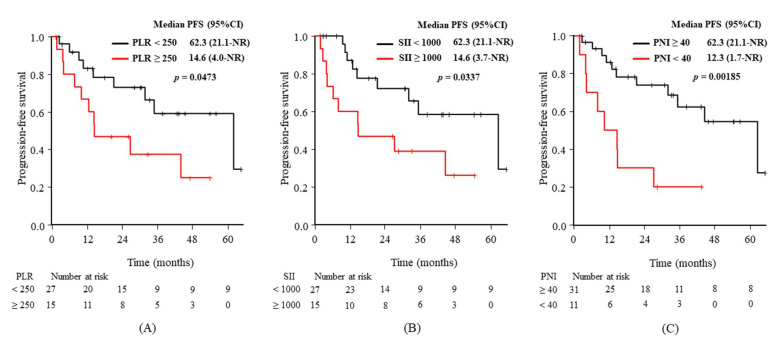
Kaplan–Meier curves of progression-free survival (PFS) according to the baseline values for (**A**) platelet-to-lymphocyte ratio (PLR), (**B**) systemic immune-inflammation index (SII), and (**C**) prognostic nutrition index (PNI). Longer median PFS values were associated with a PLR of <250, an SII of <1000, and a PNI of ≥40 at baseline. CI: confidence interval, NR: not reached.

**Figure 3 diagnostics-11-02170-f003:**
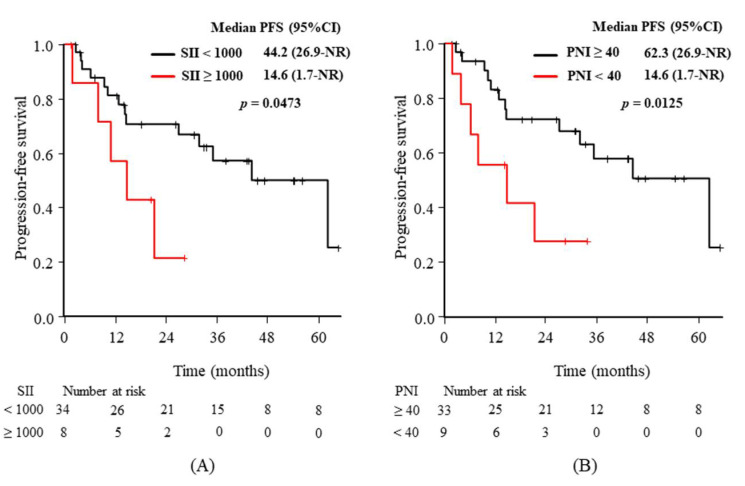
Kaplan–Meier curves of progression-free survival (PFS) according to the 3-week values for (**A**) systemic immune-inflammation index (SII) and (**B**) prognostic nutrition index (PNI). Longer median PFS values were associated with. an SII of <1000 and a PNI of ≥40 after 3 weeks of alectinib treatment. CI: confidence interval, NR: not reached.

**Figure 4 diagnostics-11-02170-f004:**
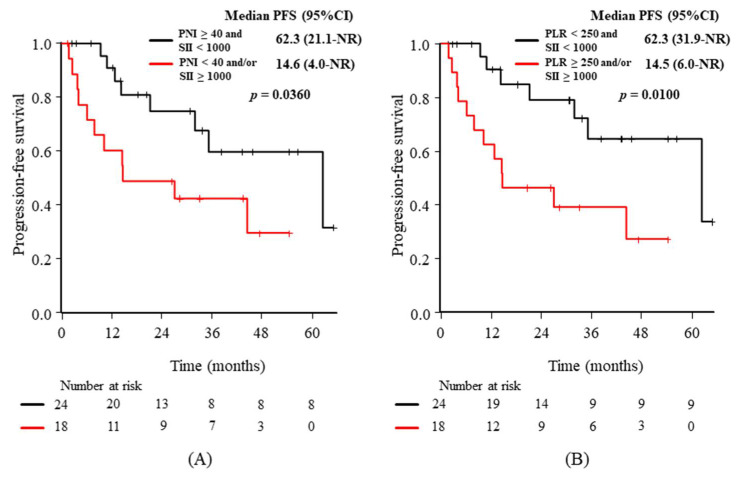
Kaplan–Meier curves of progression-free survival (PFS) according to the composite of 2 markers: (**A**) “prognostic nutrition index (PNI) ≥40 and systemic immune-inflammation index (SII) < 1000” vs. “PNI < 40 and/or SII ≥ 1000” and (**B**) “platelet-to-lymphocyte ratio (PLR) < 250 and SII < 1000” vs. “PLR ≥ 250 and/or SII ≥ 1000”. CI: confidence interval, NR: not reached.

**Table 1 diagnostics-11-02170-t001:** Baseline patient characteristics.

Characteristics	n (%)
Age (median, range)	67 years (29–85)
Sex (female/male)	20 (46.7%)/22 (52.4%)
Histology; AD/NSCLC-NOS	40 (95.2%)/2 (4.8%)
ECOG-PS; 0–1/2/3	37 (88.1%)/3 (7.1%)/2 (4.8%)
Smoking; never/current or former	12 (28.6%)/30 (71.4%)
Weight loss of ≥5%; (−)/(+)	30 (71.4%)/12 (28.6%)
**Biomarkers at baseline**	
NLR (<5 vs. ≥5)	29 (69.0%) vs. 13 (31.0%)
NLR (<3 vs. ≥3)	19 (45.2%) vs. 23 (54.8%)
PLR (<250 vs. ≥250)	27 (64.3%) vs. 15 (35.7%)
SII (<1000 vs. ≥1000)	27 (64.3%) vs. 15 (35.7%)
ALI (≥18 vs. <18)	27 (64.3%) vs. 15 (35.7%)
PNI (≥40 vs. <40)	31 (71.4%) vs. 11 (28.6%)
**Biomarkers 3 weeks after treatment**	
NLR (<5 vs. ≥5)	36 (85.7%) vs. 6 (14.3%)
NLR (<3 vs. ≥3)	27 (64.3%) vs. 15 (35.7%)
PLR (<250 vs. ≥250)	36 (85.7%) vs. 6 (14.3%)
SII (<1000 vs. ≥1000)	27 (64.3%) vs. 15 (35.7%)
ALI (≥18 vs. <18)	33 (78.6%) vs. 9 (21.4%)
PNI (≥40 vs. <40)	33 (78.6%) vs. 9 (21.4%)

AD, adenocarcinoma; ALI, advanced lung cancer inflammation index; ECOG-PS, Eastern Cooperative Oncology Group performance status; HR, hazard ratio; NLR, neutrophil-to-lymphocyte ratio; NOS, not otherwise specified; NSCLC, non-small-cell lung cancer; PLR, platelet-to-lymphocyte ratio; PNI, prognostic nutrition index; SII, systemic immune-inflammation index.

**Table 2 diagnostics-11-02170-t002:** Univariate analyses of predictors for progression-free survival (PFS).

	PFS		
	mPFS (Months, 95% CI)	HR (95% CI)	*p*-Value
Age (<70 vs. ≥70 years)	44.2 (14.6–NR) vs. 35.1 (7.8–NR)	1.29 (0.52–3.23)	0.585
Sex (female vs. male)	44.2 (21.1–NR) vs. NR (9.2–NR)	1.33 (0.53–3.32)	0.541
Smoking (− vs. +)	NR (14.4–NR) vs. 35.1 (10.1–62.3)	2.28 (0.75–6.98)	0.136
**ECOG-PS (<2 vs. ≥2)**	**44.2 (21.1–NR) vs. 10.7 (1.7–NR)**	**3.75 (1.04–13.49)**	**0.0300**
BW loss of ≥5% (− vs. +)	35.1 (14.6–NR) vs. 44.2 (6.0–NR)	1.12 (0.42–2.98)	0.827
**Before treatment**			
NLR (<5 vs. ≥5)	62.3 (14.2–NR) vs. 26.9 (4.0–NR)	1.61 (0.64–4.01)	0.307
NLR (<3 vs. ≥3)	62.3 (10.7–NR) vs. 44.2 (14.2–NR)	1.14 (0.46–2.83)	0.783
**PLR (<250 vs. ≥250)**	**62.3 (21.1–NR) vs. 14.6 (4.0–NR)**	**2.49 (0.98–6.34)**	**0.0473**
**SII (<1000 vs. ≥1000)**	**62.3 (21.1–NR) vs. 14.6 (3.7–NR)**	**2.65 (1.04–6.72)**	**0.0337**
ALI (≥18 vs. <18)	35.1 (14.2–NR) vs. 44.2 (4.0–NR)	0.98 (0.95–1.01)	0.917
**PNI (≥40 vs. <40)**	**62.3 (31.9–NR) vs. 12.3 (1.7–NR)**	**4.15 (1.57–10.9)**	**0.00185**
**[After 3 weeks]**			
NLR (<5 vs. ≥5)	44.2 (14.4–NR) vs. 17.9 (1.7–NR)	1.53 (0.49–4.77)	0.459
NLR (<3 vs. ≥3)	44.2 (26.9–NR) vs. 14.6 (4.0–NR)	1.89 (0.75–4.78)	0.174
PLR (<250 vs. ≥250)	44.2 (14.6–NR) vs. 7.8 (1.7–NR)	1.98 (0.56–7.05)	0.284
**SII (<1000 vs. ≥1000)**	**44.2 (26.9–NR) vs. 14.6 (1.7–NR)**	**2.85 (0.97–8.43)**	**0.0473**
ALI (≥18 vs. <18)	35.1 (14.2–NR) vs. 44.2 (4.0–NR)	2.00 (0.53–7.46)	0.304
**PNI (≥40 vs. <40)**	**62.3 (26.9–NR) vs. 14.6 (1.7–NR)**	**3.04 (1.09–8.46)**	**0.0125**

ALI, advanced lung cancer inflammation index; CI, confidence interval; ECOG-PS, Eastern Cooperative Oncology Group performance status; HR, hazard ratio; NLR, neutrophil-to-lymphocyte ratio; NR, not reached; PNI, prognostic nutritional index; PLR, platelet-to-lymphocyte ratio; PNI, prognostic nutrition index; SII, systemic immune-inflammation index.

**Table 3 diagnostics-11-02170-t003:** Adverse events.

	Any Grade (%)	Grade 2 (%)
Any event	10 (23.8)	2 (4.8)
Renal dysfunction	4 (9.5)	1 (2.4)
Edema	2 (4.8)	0 (0)
Peripheral neuropathy	2 (4.8)	0 (0)
Anemia	1 (2.4)	0 (0)
Bradycardia	1 (2.4)	0 (0)
Skin toxicity	1 (2.4)	0 (0)
Liver dysfunction	1 (2.4)	1 (2.4)

## Data Availability

Not applicable.

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
