# Peer review of "Prognostic Markers of Survival among Japanese Patients with Anaplastic Lymphoma Kinase-Positive Non-Small-Cell Lung Cancer Receiving First-Line Alectinib"

_diagnostics, 2021, doi:10.3390/diagnostics11122170_

Round 1

Reviewer 1 Report

Takeda et al proposed a manuscript investigating various potential prognostic markers (including NLR, PLR, SII and ALI) in 42 ALK+ NSLC treated by 1st line alectinib. 

This is an original work, not reported yet. In general, the MS is well organized, clearly presented, referenced timely and adapted. Methods are also adapted.

As 2 main points that i would suggest to add :

  • "Nutritional markers did not change in favor of better outcomes" : to support this sentence, it would be great to investigate the association between the outcomes and the VARIATION (Delta = T0 - (T+3w)) for each parameters
  • If effectives allow sub-group comparisons, it would be appreciated if 2 groups ( Patients WITHOUT any positive nutritionnal marker vs others), the same for immunological markers, and maybe 3 groups : Group A (SII- AND ALI-) vs Group B (SII+ OR ALI+) and Group C (SII+ AND ALI+). Group B and C could be merged ? Additive effects could be appreciated by this way in my point of view

Finally, i noticed some minor points : 

PFS are expressed in days. Months could help to compare with external ALK+ trials 

In Tables 'Before treatment" and "After 3 weeks" could be bold or more visual

The treatment responses in the 42 patients were classified as complete reponse in ... -> This endpoints was assessed at 3 weeks ? please mention it

Reference 1 : I think that authors meens globOcan

Thanks to the authors and the editorial team for this review request

Author Response

Response to the reviewer:

I really appreciate your sincere suggestion which is based on your profound understanding of our manuscript. I have carefully read through your comments, and our responses are given below in a point-by-point manner.

As 2 main points that i would suggest to add :

  • "Nutritional markers did not change in favor of better outcomes" : to support this sentence, it would be great to investigate the association between the outcomes and the VARIATION (Delta = T0 - (T+3w)) for each parameters

Response:

 I really appreciate your sincere suggestion and advice. As you pointed out, using the difference between the 3-week value and the baseline value would be more effective in describing the trend of the relevant markers during alectinib treatment. Then, we assessed the delta (= [T+3W] - T0), and 10% change in SII, ALI, and PNI was defined as significant. Since PLR showed distribution with large deviation, 30% change in PLR was defined as significant. However, we found no statistically significant difference: p-values for SII, ALI, PNI, and PLR were 0.449, 0.841, 0.079, and 0.449, respectively.

 Since there were too many parameters, we did not add this description in the current manuscript.

  • If effectives allow sub-group comparisons, it would be appreciated if 2 groups ( Patients WITHOUT any positive nutritionnal marker vs others), the same for immunological markers, and maybe 3 groups : Group A (SII- AND ALI-) vs Group B (SII+ OR ALI+) and Group C (SII+ AND ALI+). Group B and C could be merged ? Additive effects could be appreciated by this way in my point of view

Response:

 I really appreciate your sincere suggestion. As you mentioned, subgroup analysis by the combination of positive markers could lead to the creation of more sensitive composite markers. As you mentioned, I agree with your suggestion of analyzing 2 groups rather than 3 groups, considering the small sample size.

I understood that “(-)” by your definition meant the favor population according to the relevant markers, since you intended to get additive effect by combining two markers. We chose one of the nutritional markers (ALI or PNI) and one of the immunological markers (PLR or SII). Furthermore, I added a composite of statistically significant immunological markers (PLR and SII).

Therefore, we additionally analyzed the followings: (1) “ALI ≥ 18 and PLR < 250” vs. “ALI < 18 and/or PLR ≥ 250”, (2) “ALI ≥ 18 and SII < 1000” vs. “ALI < 18 and/or SII ≥ 1000”, (3) “PNI ≥ 40 and PLR <250” vs “PNI < 40 and/or PLR ≥ 250”, (4) “PNI ≥ 40 and SII < 1000” vs. “PNI < 40 and/or SII ≥ 1000”, (5) “PLR < 250 and SII < 1000” vs. “PLR ≥ 250 and/or SII ≥ 1000”.

Although the additive effects were expected through the process, the statistically significant findings were observed in (4) and (5). P-values for abovementioned 5 groups were 0.475 (1), 0.365 (2), 0.181 (3), 0.036 (4), and 0.010 (5), respectively.

Therefore, we adopted the combination of “PNI and SII” and “PLR and SII” as the potential composite markers. We added these additional analyses in “Study variables” subsection of Patients and Methods section, “Relationship between the baseline characteristics or candidate predictive biomarkers and PFS” subsection of the Results section, and in Figure 4.

I did not add the discussion on this additional analysis in the Discussion section, since the addition would make the discussion redundant. If you think the addition is important, I am willing to enrich the Discussion section.

Finally, i noticed some minor points : 

PFS are expressed in days. Months could help to compare with external ALK+ trials 

Response:

 I really appreciate your suggestion. I changed the relevant parts into months.

In Tables 'Before treatment" and "After 3 weeks" could be bold or more visual

Response:

 I appreciate your advice. When I changed the relevant parts into bold, it became difficult to distinguish them from bold parts such as ECOG-PS, PLR, SII, and PNI in Table 2. Then I enlarged the font size and used brackets as well as changing them into bold to highlight “Before treatment” and “After 3 weeks”.

The treatment responses in the 42 patients were classified as complete reponse in ... -> This endpoints was assessed at 3 weeks ? please mention it

Response:

 I thank you for pointing out the timing of evaluation. The first evaluation was conducted after 3 weeks. Then, I added “evaluated after 3 weeks of alectinib treatment” in the “ORR and PFS outcomes” subsection of the Results section.

Reference 1 : I think that authors meens globOcan

Response:

 I thank you for your comment. As you indicated, the reference No.1 meant Global cancer statistics 2018: GLOBOCAN estimates.

Reviewer 2 Report

Authors designed a multicenter retrospective study to evaluate the immunological and nutritional markers at baseline and 3 weeks 24 after alectinib initiation, and showed that PFS outcomes were 166 significantly associated with baseline ECOG-PS (hazard ratio [HR]: 3.75, p-value: 0.0300), PLR (HR: 2.49, p-value:0.0473), SII (HR: 2.65, p-value: 0.0337), and PNI (HR: 4.15, p-value: 0.00185), as well as 3-week values for SII (HR: 2.85, p-value: 0.0473) and PNI (HR: 3.04, p- value: 0.0125).

There were several concerns about the manuscript.

  1. Though authors claimed they evaluate the immunological and nutritional markers, PLR, PNJ, SII and PNI cannot represent as reliable markers in such studies. For example, in stage IV NSCLC patients harbored ALK, there were many clinical conditions can interfere the neutrophil, lymphocytes and platelets, such as viral infection or bacterial infection, autoimmune diseases, disseminated intravascular coagulopathy and so on. These terminal stage NSCLC patients have to receive Alectinib urgently in conditions with many confounding factors, the sceneries made the markers not reliable.
  2. Since authors announced the study is to evaluate the prognostic factors, and the enrolled patients sample were very small, authors have to collect more information such as diagnostic criteria (IHC or RT-PCR or FISH or NGS), TNM and stage, metastasis sites (brain, liver, bone, and so on..), ALK variants (V1-V3).
  3. In page 3 “ Baseline characteristics namely; age, sex, smoking history, ECOG-PS score, comorbidities, height, body weight, weight loss of ≥ 5% 6 months before alectinib initiation, body mass index (BMI), histology, blood examination, and the efficacy of alectinib” . However, no any comorbidities were mentioned and calculated in the study, and blood tests, such as albumin, CRP, ESR(inflammation markers), liver or renal function, were also not included in the analysis of the study.
  4. The dosage of Alectinib in these enrolled patients is lacking. All patients received the same dosage, 300mg bid or 600mg bid?. The dosage may interfere the outcome as the data in J-Alex and Alex trial. In addition, 2 patients experienced grade 2 ADR, did any patient need to discontinue or reduced dosage of Alectinib?

Author Response

Response to the Reviewer:

I really appreciate your sincere advice and critique, and respond to your suggestions in a point-by-point manner below.

1. Though authors claimed they evaluate the immunological and nutritional markers, PLR, PNJ, SII and PNI cannot represent as reliable markers in such studies. For example, in stage IV NSCLC patients harbored ALK, there were many clinical conditions can interfere the neutrophil, lymphocytes and platelets, such as viral infection or bacterial infection, autoimmune diseases, disseminated intravascular coagulopathy and so on. These terminal stage NSCLC patients have to receive Alectinib urgently in conditions with many confounding factors, the sceneries made the markers not reliable.

Response:

 I agree with you in that the constituents of the current immunological and nutritional markers are easily affected by several conditions other than lung cancer.

And I also agree with you in that alectinib treatment should be initiated irrespective of the proposed immunological and nutritional markers when the relevant patients meet indication criteria.

 However, I believe that the above-mentioned matters would never spoil the importance of these immunological and nutritional markers during cancer treatment, since these markers have been investigated as prognostic markers in several cancers as far as anticancer treatment is acceptable. Therefore, I think that our retrospective study should be conducted in accordance with the historical investigation, not excluding other specific factors.

2. Since authors announced the study is to evaluate the prognostic factors, and the enrolled patients sample were very small, authors have to collect more information such as diagnostic criteria (IHC or RT-PCR or FISH or NGS), TNM and stage, metastasis sites (brain, liver, bone, and so on..), ALK variants (V1-V3).

Response:

 I thank you for your important advice. I added the diagnostic methods of ALK rearrangement in the “Patients” subsection of the Patients and Methods section. The diagnostic procedures were IHC in 11 patients, IHC and FISH in 27, IHC and RT-PCR in 1, FISH in 1, RT-PCR in 1, and NGS (Oncomine Dx Target Test Multi-CDx system) in 1 patient. ALK variants are not routinely investigated in Japan.

 The additional investigation according to TNM, clinical stage, metastatic sites seem to make our manuscript more complex and difficult to understand the outcome, considering the small sample size. Then, I added an expression “metastatic or recurrent ALK-positive NSCLC” in the “Patients” subsection of the Patients and Methods section, in order to clarify the status of the participants.

3. In page 3 “ Baseline characteristics namely; age, sex, smoking history, ECOG-PS score, comorbidities, height, body weight, weight loss of ≥ 5% 6 months before alectinib initiation, body mass index (BMI), histology, blood examination, and the efficacy of alectinib” . However, no any comorbidities were mentioned and calculated in the study, and blood tests, such as albumin, CRP, ESR(inflammation markers), liver or renal function, were also not included in the analysis of the study.

Response:

 I thank you for your advice. As you mentioned, the comorbidities were not presented in the manuscript. Then, I deleted this term in the “Data collection” subsection of the Patients and Methods section. The data of blood examination were used to calculate the immunological and nutritional markers. Then, we had to collect these data. I think that the analysis of each value is not important in the current study.

4. The dosage of Alectinib in these enrolled patients is lacking. All patients received the same dosage, 300mg bid or 600mg bid?. The dosage may interfere the outcome as the data in J-Alex and Alex trial. In addition, 2 patients experienced grade 2 ADR, did any patient need to discontinue or reduced dosage of Alectinib?

Response:

 I appreciate your important suggestion and agree with your opinion. All patients received alectinib at a dose of 300mg twice daily. I added this description in the “Patients” subsection of the Patients and Methods section. Those patients who developed grade 2 renal dysfunction due to transient dehydration, and liver dysfunction did not need dose reduction. I added these descriptions in the “Adverse events” subsection in the Results section.

Reviewer 3 Report

The present manuscript reports on a retrospective study evaluating different immunological and nutritional markers as prognostic markers during treatment with alectinib.

Some comments:

  • Median PFS should be indicated in months, as it is more commonly reported in this form. Otherwise, the readers might be confounded.
  • The authors should better clarify the selected cut-off value for PLR. Previous studies have, for instance used 200 as cut-off (Russo A, et al. Adv Ther 2020) or tertiles (Diem S, et al. Lung Cancer 2017). Please clarify
  • Recently, the LIPI score has proved to be prognostic in NSCLC (Kazandijan D, et al. JAMA Oncol 2019). The authors should explain why they did not include this parameter.
  • Other studies have recently reported the clinical utility of dNLR dynamics analysis in patients treated with immunotherapy (Mezquita L, et al. Eur J Cancer 2021).

Author Response

Response to the Reviewer:

I really appreciate your profound understanding of our manuscript as well as sincere suggestion and advice in order to improved our manuscript. I have carefully read your comments and our responses are given below in a point-by-point manner.

Some comments:

  • Median PFS should be indicated in months, as it is more commonly reported in this form. Otherwise, the readers might be confounded.

Response:

 I really appreciate your suggestion and I have changed the relevant parts into months.

  • The authors should better clarify the selected cut-off value for PLR. Previous studies have, for instance used 200 as cut-off (Russo A, et al. Adv Ther 2020) or tertiles (Diem S, et al. Lung Cancer 2017). Please clarify

Response:

 I appreciate your important suggestion and critique. This cut-off value for PLR was defined according to the previous report (Ref.18: Oncol Lett. 2016;11(3):2241-8.), where the cut-off value for PLR was set at 246.36. Then, I added the reason why we adopted this cut-off in the “Study variables” subsection of the Patients and Methods section.

  • Recently, the LIPI score has proved to be prognostic in NSCLC (Kazandijan D, et al. JAMA Oncol 2019). The authors should explain why they did not include this parameter.

Response:

 I appreciate your sincere suggestion. We did not adopt the LIPI because the LIPI exhibited association with immune checkpoint inhibitor treatment, while failing to show significant association with chemotherapy. Alectinib as a molecular targeting agent would not be suitable for the LIPI analysis. Thus, we did not include LIPI in the submitted manuscript.

However, we conducted an analysis of LIPI according to your suggestion, and there was no significant association between LIPI and median PFS (p=0.357).

 I did not adopt LIPI this time so as not to confuse the potential readers with too many parameters.

 Then, I added the reason behind the absence of LIPI analysis in “the Limitation” of the Discussion section in the revised manuscript, adding the following expression: “Thirdly, we selected the candidate immunological and nutritional markers based on the results of the original articles. For example, we did not adopt lung immune prognostic index (LIPI), because LIPI was associated with the outcomes of immune checkpoint inhibitors but not with chemotherapy [42]. Therefore, there could be some potential markers which we did not analyze.”

 However, if you think disclosing the negative data would be important, I am willing to add them in the manuscript.

  • Other studies have recently reported the clinical utility of dNLR dynamics analysis in patients treated with immunotherapy (Mezquita L, et al. Eur J Cancer 2021).

Response:

 I thank you for your important suggestion. We adopted NLR with two cut-off values (NLR <5 vs ≥5, <3 vs ≥3) and in two points in time (baseline and after 3 weeks) in the current manuscript. The derived NLR is almost the same with NLR, and we selected NLR. The dNLR dynamics is interesting, however, the number of markers had become too many and we avoided the confusion.

We conducted dNLR analysis according to your suggestion and no significant association was observed (p=0.611). Then, I decided not to adopt dNLR analysis including dNLR dynamics in the current study. However, if you think disclosing these negative data would be important, I am willing to add them in the manuscript.

Round 2

Reviewer 2 Report

The manuscript has been much improved in current form, and accept could be considered.